# REPRESENTATION LEARNING FOR REMOTE SENSING: AN UNSUPERVISED SENSOR FUSION APPROACH

## ABSTRACT

In the application of machine learning to remote sensing, labeled data is often scarce or expensive, which impedes the training of powerful models like deep convolutional neural networks. Although unlabeled data is abundant, recent self-supervised learning approaches are ill-suited to the remote sensing domain. In addition, most remote sensing applications currently use only a small subset of the multi-sensor, multi-channel information available, motivating the need for fused multi-sensor representations. We propose a new self-supervised training objective, Contrastive Sensor Fusion, which exploits coterminous data from multiple sources to learn useful representations of every possible combination of those sources. This method uses information common across multiple sensors and bands by training a single model to produce a representation that remains similar when any subset of its input channels is used. Using a dataset of 47 million unlabeled coterminous image triplets, we train an encoder to produce semantically meaningful representations from any possible combination of channels from the input sensors. These representations outperform fully supervised ImageNet weights on a remote sensing classification task and improve as more sensors are fused. Our code is available at https://storage.cloud.google.com/public-published-datasets/csf_code.zip.

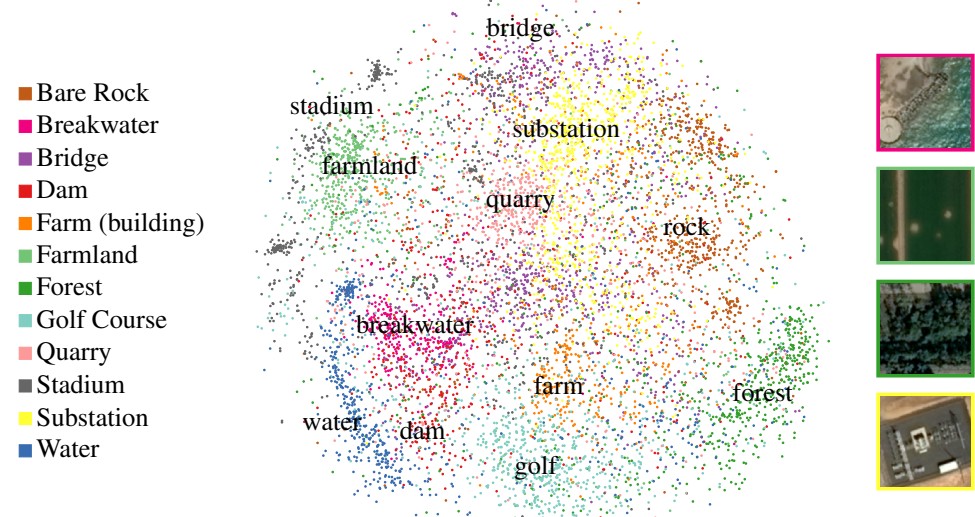

Figure 1: Learned representations of out-of-sample image scenes, visualized with PCA followed by t-SNE and colored by OpenStreetMap category. Without any labels, Contrastive Sensor Fusion has learned a representation that groups remote sensing images into semantically meaningful categories.

## 1 INTRODUCTION

Remote sensing data has become broadly available at the petabyte scale, offering unprecedented visibility into natural and human activity across the Earth. Many techniques have been recently

developed for applying this data with machine learning to solve geospatial tasks like semantic segmentation (Audebert et al., 2016), (Kampffmeyer et al., 2016), broad-area search (Keisler et al., 2019), and classification (Maggiori et al., 2016), (Sherrah, 2016).

Due to the complexity and visuospatial nature of solving problems with aerial imagery, it is natural to use deep convolutional neural networks, but CNNs typically require large amounts of labeled data to achieve good performance. In remote sensing, these labels are usually scarce and hard to obtain; semantic segmentation requires boundaries to be labeled at single-pixel precision. A modern approach to the problem of data scarcity is semi-supervised learning, which uses unlabeled data to ease the task of learning from small amounts of labeled data. This approach is particularly well-suited to remote sensing because of the amount of unlabeled data available.

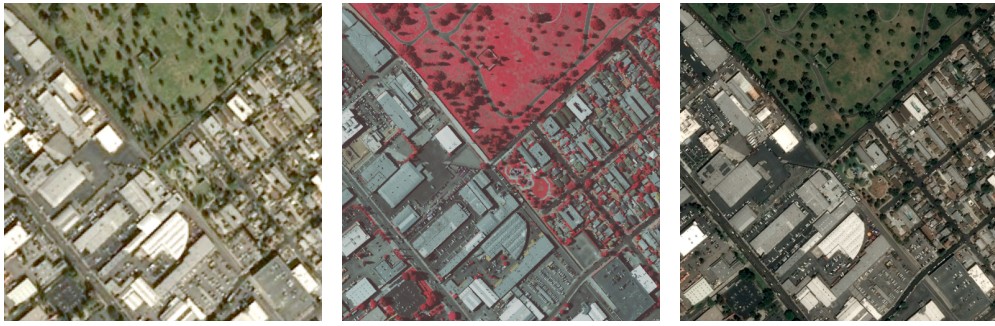

Figure 2: Coterminous remote sensing imagery from three different sensors: Airbus SPOT, NAIP (visualized here in near-infrared, red, and green), and Airbus Pléiades (see Appendix A.3 for details). As seen here, images contain many small components (roads, buildings, structures, trees) and adjacent locations can look completely different (e.g., the transition from buildings to grass). We leverage these multiple views to generate representations with any subset of available sensors or channels.

While most self-supervised and unsupervised image analysis techniques focus on natural imagery, remote sensing differs in several critical ways, requiring a different approach. Where pictures taken by a human photographer often have one or few subjects, remote sensing images like those in Figure 2 contain numerous objects such as buildings, trees, or factories. Additionally, the important content can change unpredictably within just a few pixels or between images at the same location from different times. Multiple satellite and aerial imaging platforms capture images of the same locations on earth with a wide variety of resolutions, spectral bands (channels), and revisit rates, such that any specific problem can require a different combination of sensor inputs (Reiche et al., 2018; Rustowicz et al., 2019).

Recent research in semi-supervised learning has led to a wealth of methods that achieve success on problems like classifying natural images (van den Oord et al., 2018; Tian et al., 2019; Hénaff et al., 2019) and understanding language (Mikolov et al., 2013; Devlin et al., 2018). These approaches almost universally rely on the "distributional hypothesis": the property that parts of the data that are close in time or space are similar. Previous work (Jean et al., 2019) has learned representations for overhead imagery using the distributional hypothesis. However, we argue that it is less applicable in remote sensing, due to the aforementioned differences between overhead imagery and these domains. The distributional hypothesis still applies, but the scale of correlated patches is smaller, making it difficult to harvest scenes for related patches. We therefore modify these techniques to build on the intuition that similar layouts of objects on the ground should have similar representations, regardless of the combination of sensors.

In this paper, we use this idea to develop a method of learning representations for overhead imagery, which we call Contrastive Sensor Fusion (CSF). We train a single model, unsupervised, to produce a representation of a scene given any subset of the sensors used during training over that scene. During training, we form two "views" of each scene from a small random subset of the available sensor channels (channel/sensor dropout). Then both views are encoded with the same network and the resulting representations are compared, using a contrastive loss to encourage the two views to have similar representations.

This model is trained on a ~20 TB dataset consisting of 47 million scenes, each consisting of four bands for each of three different sensors over that scene. We perform several experiments, showing that the unsupervised encoder learns to produce representations of many different combinations of sensors that separate images into semantically-meaningful classes and perform well on real problems in the semi-supervised setting. The representations are transferred as-is, with no fine-tuning step or stacking additional models.

## 2 BACKGROUND

Contrastive methods train models by using the representation of one observation to predict the representation of a different but related observation. Examples include pairing frames from the same video (Wang and Gupta, 2015) or words from the same local context (Mikolov et al., 2013). The recent method Contrastive Predictive Coding (CPC) uses a representation of the current time step in a sequence to predict the same encoder's representation of a future time step (van den Oord et al., 2018; Hénaff et al., 2019). CPC has been shown to learn expressive representations of natural imagery by using the representation of one image sub-patch to predict the representation of adjacent sub-patches, contrasting against patches drawn from other images. However, this adjacent-patch technique fails for remote sensing imagery, which can change abruptly between adjacent patches.

Contrastive Multiview Coding (CMC, Tian et al. (2019)) is a related approach that compares multiple views of the same context. Unlike CPC, CMC does not assume that representations of image patches should be stable in space and benefits from having many views of a scene, making it applicable to remote sensing. On the other hand, CMC relies on a fixed partitioning of data sources, forcing practitioners to decide ahead of time which sensor combinations to fuse. It also trains a separate model for each view, limiting how much information can be shared between views.

We use the InfoNCE loss introduced by van den Oord et al. (2018). Intuitively, its role is to train the encoder to make representations of the same scene similar across views. Computing the loss at the level of *representations* instead of *pixels* sidesteps issues with reconstruction-based loss functions like autoencoders use, which are dominated by low-level properties of pixel space like overall brightness instead of high-level information; however, directly predicting one representation from another leads to trivial solutions, such as representing every scene with the zero vector. Instead, contrastive losses like InfoNCE ask the network to classify which representation among a set including many "noise examples" belongs to the same scene.

We borrow a number of ideas directly from Deep InfoMax (Hjelm et al., 2019) and its successor, Augmented Multiscale Deep InfoMax (AMDIM) (Bachman et al., 2019). These methods learn by contrasting representations at various layers and spatial locations of the same network. AMDIM increases the difficulty of this task by independently augmenting each view. Computing the loss at multiple levels and sharing the network weights between views also increase supervisory signal to the model. As with CPC, however, AMDIM assumes that distinct patches from the same image are closely related, which is problematic in remote sensing.

Finally, we tested the non-contrastive method of Split-Brain Autoencoders (Zhang et al., 2017), but found them to be less effective than contrastive methods. Other pixel-space methods (e.g. Singh et al. 2018) exist for remote sensing representation learning, but these face the issues with reconstruction-based losses mentioned above.

## 3 CONTRASTIVE SENSOR FUSION

We introduce Contrastive Sensor Fusion (CSF, Figure 3), a technique for learning unsupervised representations of every combination of its input sensors. CSF learns by creating two views of each location by randomly sampling two sensor combinations, encoding both views with the same CNN (sharing weights), and comparing representations across locations with a contrastive loss that aligns representations of the same location regardless of the input sensor combination.

Our method can be seen as an extension of Contrastive Multiview Coding, with three critical differences. First, instead of dividing the input channels into fixed groups during training, CSF drops out channels at random, forcing the network to learn to encode every combination of channels. Second, CSF uses a "Siamese network" (Bromley et al., 1993) training scheme, sharing weights

across the encoders of every sensor combination rather than training a separate encoder for each view. Third, CSF computes a contrastive loss at multiple layers of representation, which helps to learn localized representations. In a sense, CSF can be seen as a computationally tractable way of training an ensemble of all exponentially-many[1] splits of views in CMC at once, with weight sharing between each encoder and loss at multiple levels. This is analogous to the extension of NADE models (Larochelle and Murray, 2011) to ensembles of arbitrary ordering (Uria et al., 2014).

## 3.1 LEARNING FROM MULTIPLE VIEWS

Like CMC, CSF learns by contrasting representations of multiple views of the same input. Given a set of sensor looks $X$ over a scene, we aim to create two views $V^{(1)}(X), V^{(2)}(X)$ that contain the same high-level information (e.g. object identities and location, scene type, topography) but differ as much as possible at the pixel level. In order to match representations of the same scene, the model must encode both views in such a way that the scene remains distinctive. If the views differ enough, this forces the encoder to capture high-level information about the scene, since that is the most salient remaining information common to both views, while discarding nuisance factors like lighting, resolution, and differences between sensors.

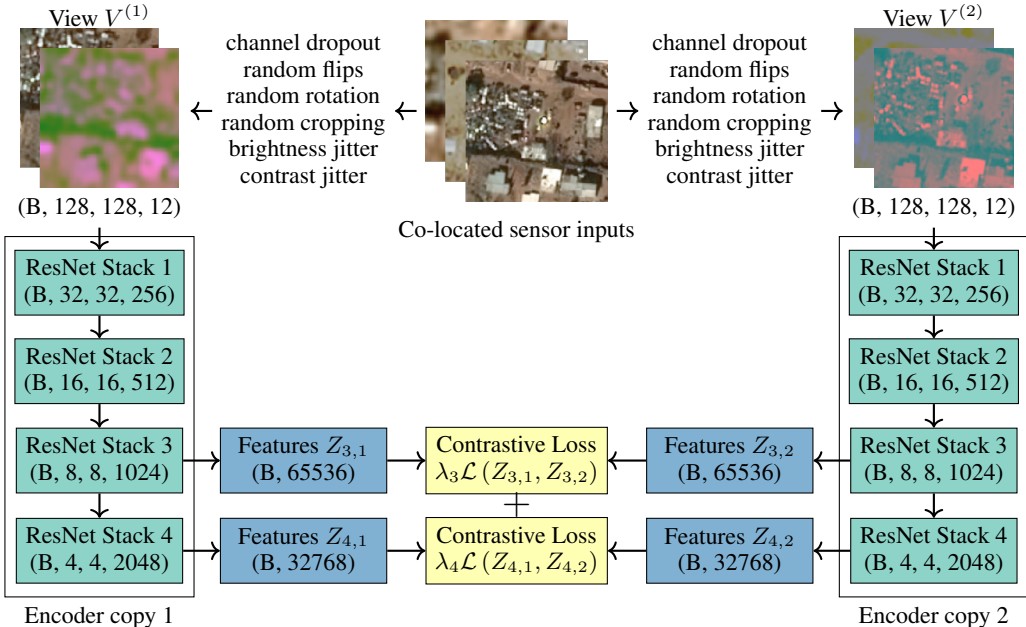

Figure 3: Contrastive Sensor Fusion architecture during training. Weights are shared across encoder copies. The contrastive loss trains the encoder to represent the same (different) location the same (different) way regardless of sensor/channel combination. The process to create views is explained in more detail in Figure A.6, and the computation of the loss is detailed in Appendix A.2.

To create a view $V(X)$, we randomly set a large fraction of the channels in $X$ to zero, which effectively fuses a different combination of sensor channels for each example. With a channel dropout rate of $p$, we scale the remaining channels by $\frac{1}{1-p}$ as in Srivastava et al. (2014). Similarly, at inference time we run the encoder with any subset of the channels it was trained with, replacing the others with zeros and scaling up remaining channels by the same factor as was used during training.

We augment images by randomly cropping a small number of pixels from $X$, applying random flips and rotations, and jittering the brightness and contrast of each remaining channel; these increase the difficulty of the contrastive task and improve the robustness of the representation to nuisances.

---

[1] Given $n$ sensors, there are $2^n - 1$ possible sensor combinations (counting all subsets and discounting the empty set), and so as $n$ grows, training a separate model for each is infeasible.

Because CSF uses a single network for multiple sensors with potentially different resolutions, numbers of channels, and even modalities, care must be taken to design an encoder which can fuse them effectively. In our experiments, we used three sources of optical imagery with slightly different pixel resolutions (see A.3). We bilinearly upsampled[2] the two lower-resolution sources to match the highest resolution, then concatenated the sensors' channels depth-wise such that each pixel fed to the first convolution covers the same physical area.

## 3.2 InfoNCE Loss

CSF compares representations at multiple levels of the encoder using the InfoNCE loss. As van den Oord et al. (2018) show, InfoNCE maximizes the mutual information (MI) between representations. We design our cross-view augmentation to destroy MI between views except for the high-level understanding, so maximizing the MI between representations directly trains the encoder to learn robust and expressive representations of its input.

At each of several layers $L$ in the encoder, a contrastive loss is computed as follows. Let $V$ be a stochastic function which maps a set of sensor looks $X$ to a view, and let $Z_L$ be a function which maps a view to the encoder's layer-$L$ representation of that view. We apply $V$ to $X$ twice to produce two views and encode both into layer-$L$ representations, which we denote by $Z_L^{(1)} \equiv Z_L(V^{(1)}(X))$ and $Z_L^{(2)} \equiv Z_L(V^{(2)}(X))$. Let $\phi$ be a scoring function on this representation space. $\phi$ plays the role of contrastive learning's predictive model; given two views, it produces the logit of the two views belonging to the same scene. Given a collection of $n$ other scenes $X_{\text{noise}} = \{\tilde{X}^1, \ldots \tilde{X}^n \neq X\}$ and their layer-$L$ representations $Z_{L,\text{noise}} = \{Z_L(V(\tilde{X}^1)), \ldots Z_L(V(\tilde{X}^n))\}$ used as noise examples to classify among, the loss at layer $L$ for example $X$ is the InfoNCE loss presented in CPC,

$$\mathcal{L}_L^{\text{forward}}(X) = -\log \frac{\exp(\phi(Z_L^{(1)}, Z_L^{(2)}))}{\exp(\phi(Z_L^{(1)}, Z_L^{(2)})) + \sum_{\tilde{Z} \in \tilde{Z}_{L,\text{noise}}} \exp(\phi(Z_L^{(1)}, \tilde{Z}))}$$

This is the loss function for predicting view 2 from view 1, which we refer to as the "forward" loss. Because the problem is symmetric (we are comparing representations of the same encoder run over two identically-created views), we can use the same procedure and set of negatives to predict view 1 from view 2, the "backward" loss. The total contrastive loss for layer $L$, $\mathcal{L}_L$, is the sum of the two, and the total loss for example $X$ is

$$\mathcal{L}_{\text{tot}} = \sum_L \lambda_L \mathcal{L}_L$$

where $\lambda_L$ weights the contrastive loss at layer $L$. Following the intuition that two views of a scene are most similar in abstract high-level ways, we use nonzero weight on only last layers of the last two residual stacks, and weigh the last layer twice as heavily.

In all experiments, we use $\phi(Z_L^{(1)}, Z_L^{(2)}) = Z_L^{(1)^T} Z_L^{(2)}$ and draw negative examples from other scenes in the batch. This allows very efficient computation of the contrastive loss. See A.2 for more details.

Computing contrastive loss at multiple layers is efficient, since it requires applying the encoder only once per view, and is beneficial for a number of reasons. Since modern CNNs use multiple layers of pooling, higher layers in the encoder are naturally more translation invariant and so multi-layer loss helps CSF learn well-localized features useful for segmentation. Additionally, the extra supervisory signal helps earlier layers learn more quickly.

## 3.3 Training

During training, both copies of the encoder share weights. As argued above, this trains a single model which can function with any of its input channels missing. In addition, weight-sharing makes the encoder more parameter-efficient. This can lead to information sharing across sensors; for example, the network may learn to combine the blue channel of one sensor with the green channel of another early in the network, since the two bands often provide similar information.

---

[2] For sensors of very different resolutions, training a separate head for each sensor before combining the representations may be appropriate.

In contrast to CPC and AMDIM, computing the loss does not require masking the encoder or using an encoder with a restricted receptive field size. CSF learns representations of whole images and contrasts representations from distinct scenes rather than distinct parts of the same image, so a standard architecture can be used for the encoder. Following van den Oord et al. (2018), we use a ResNet encoder (He et al., 2015). We choose ResNet-50 instead of ResNet-101 for our experiments because its smaller memory footprint enables us to use a larger batch size. Larger batch sizes result in more accurate loss since the negative examples for NCE are the other scenes in the batch. We train on a single TPU v2 with a batch size of 2048.

During training, we schedule the channel dropout rate to increase linearly from 0 to 0.66 over the first 8000 batches. This is a form of curriculum learning (Bengio et al., 2009) where the contrastive task starts in an easier setting where more band are retained, and gets gradually harder throughout training; we observed faster convergence with this channel dropout schedule. We also linearly warm up the learning rate over the first 3000 batches (Goyal et al. 2017). Other hyperparameters are kept constant; when making each view, we randomly crop away 32 pixels and randomly jitter the brightness and contrast of each retained band by at most 25%.

## 4 EXPERIMENTS

We test this method by training a CSF network on 47 million image triples from three imaging platforms: Airbus SPOT, USDA NAIP, and Airbus Pléiades, with pixel resolutions of 150 cm, 100 cm, and 50 cm, respectively. Each has four channels: red, green, blue, and near-infrared (NIR). See Appendix A.3 for details. To visualize the feature space and evaluate the learned representations, we built a dataset of 8400 samples based on OpenStreetMap (OSM) features from 12 classes of distinctive objects; example images are shown in Figure 1 and Figure 5. Each object is seen in all three sensors used to train the encoder. See Appendix A.4 for details.

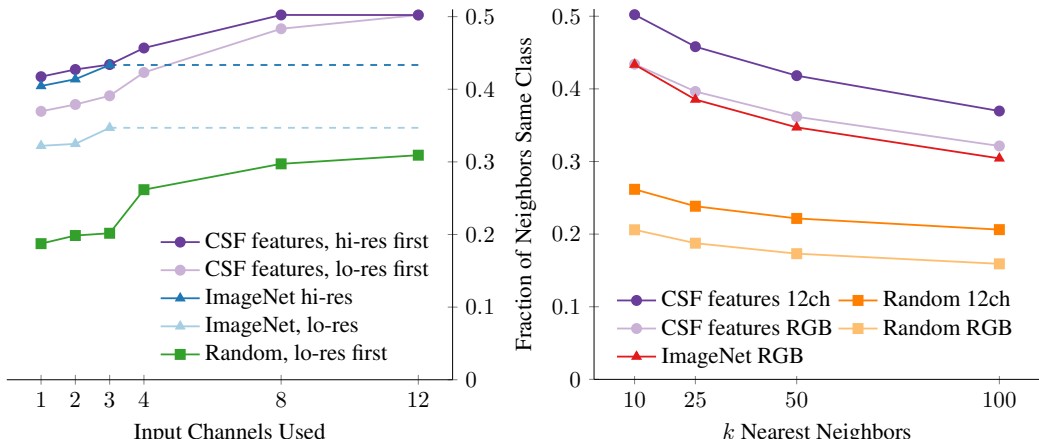

Figure 4:   We compare the clustering of features based on OSM class using a nearest neighbor metric. The plots show the fraction of same-class neighbors for each point ($k = 10$) as input channels are added (**left**), and the fraction of same-class neighbors as a function of $k$ (**right**). One, two, and three-channel experiments always use a single sensor, taking the red band only, the red and green bands, and the RGB bands respectively. Our features outperform ImageNet's in this unsupervised clustering metric and improve when multiple sensors are fused.

We test the performance of the trained CSF network with several experiments. For each scene in the OSM-based dataset, we produce representations from a ResNet50V2 encoder initialized with weights either learned through CSF, from fully-supervised ImageNet classification, or set randomly. We use various combinations of input channels and sensors (although ImageNet weights are limited to 3 input channels). We reduce the dimensions of the generated representations by principal component analysis (PCA). In Figure 1, we visualize the CSF representations by embedding the 200 principal components of the features obtained from 12 input channels into two dimensions with t-SNE. Next, we

measure the quality of clustering by identifying the $k$-nearest neighbors to each point in representation space (2048-component PCA) and count the fraction that belong to the OSM class of that point. We also test classification performance by training a $k$-nearest neighbors (KNN) classifier with $k = 10$ and leave-one-out cross validation on every point and measure the classification accuracy (Table 1). We report the average of each metric over the OSM dataset.

Experiment results from several encoder-channel combinations are presented in Figure 4 as a function of non-zero channels (panel 1) or choice of $k$ (panel 2). Because ImageNet weights only allow three or fewer input bands, we test ImageNet weights applied to either the highest resolution (Pléiades) or lowest resolution (SPOT) RGB imagery. $k$-NN classifier accuracy is presented in Table 1.

### 4.1 REPRESENTATION QUALITY

First, we evaluate the quality of learned representations by considering how well they cluster scenes by OpenStreetMap object category. Plotting the resulting points colored by OSM label (Figure 1), we observe that most classes form distinctive clusters. Neither the encoder nor the dimensionality-reduction algorithms have access to these labels; therefore, we can observe directly that CSF clusters images into semantically meaningful categories.

Table 1: We compare the representation quality of CSF and ImageNet weights for various sensor/channel combinations by training a KNN classifier on our OSM dataset. This table shows the fraction of 10-nearest neighbors belonging to the same class, and the accuracy of the classifier.

| Weights | Channels | 10-NN same-class (%) | Accuracy (%) |
|---|---|---|---|
| ImageNet | Airbus SPOT RGB | 34.69 | 47.96 |
| | Airbus Pléiades RGB | 43.34 | 57.41 |
| CSF | Airbus SPOT RGB | 39.10 | 53.16 |
| | Airbus SPOT RGB + near-IR | 42.31 | 55.71 |
| | Airbus Pléiades RGB | 43.40 | 57.47 |
| | Airbus Pléiades RGB + near-IR | 45.68 | 60.23 |
| | All sensors, RGB + near-IR | **50.22** | **64.06** |

The clustering quality is quantified in Figure 4 and Table 1. In every case for both nearest-neighbor class and classification accuracy, representations learned through CSF outperform those learned with all ImageNet labels (Figure 4). We emphasize that training the ImageNet encoder required 14 million labels whereas the CSF encoder was trained purely unsupervised. Though overhead imagery differs significantly from ImageNet's labeled natural images, out-performing ImageNet is not trivial; previous work (Singh et al., 2018) demonstrates that supervised ImageNet pretraining is a strong baseline for representations of remote sensing imagery [3].

Furthermore, as we add channels the unsupervised CSF encoder continues to improve. We also investigated all other orders of adding sensors, and chose to display the order which resulted in lowest scores first; that is, adding the lower resolution data before the higher resolution data. This demonstrates a limitation of transfer learning from datasets like ImageNet: the resulting weights can only be used with channels present in the dataset. Most labeled data is RGB, so supervised transfer learning prevents us from taking advantage of multiple co-registered data sources.

### 4.1.1 MOTIVATION

This OSM classification task demands that a representation understand the kinds of objects and relationships useful for a broad range of remote sensing applications. It demonstrates that the features learned, whether high-level or low-level, are features which allow discriminating between semantically distinct sets of input images. Intuitively, representations which do not cluster well among semantic classes require learning more complex transformations to separate the classes.

---

[3]We compare results on a multiclass classification problem, so we expect supervised training with ImageNet labels to be particularly effective here relative to e.g. a segmentation task.

Fine-tuning experiments have the potential to demonstrate this fact, and we note this as a future avenue for research. However, we believe that special care will need to be made to show that features continue to generalize to tasks where labeled data is not available for fine-tuning.

## 4.2 SENSOR FUSION

Next, we investigate our main claim: that CSF learns to fuse different sensors with minimal degradation in feature quality when only a subset of sensors are available. We observe that clustering quality increases monotonically as channels are added (first panel of Figure 4). Though the biggest gains come from using high-resolution imagery, fusing multiple low-resolution sensors noticeably improves representations above having a single higher-resolution sensor. We were surprised to discover that fusing SPOT and NAIP imagery with CSF results in higher performance than CSF representations of (higher-resolution) Pléiades imagery, as seen by comparing the light-purple point at 8 input channels to dark-purple points at $\leq 4$ input channels. In the second panel of Figure 4, the fully fused 12-channel CSF network outperforms 3-band ImageNet at all values of $k$.

In every case, performance increases as more channels added, regardless of order. This demonstrates that CSF is effectively combining the information present across multiple sensors, and that CSF has learned a good representation for each of its possible views. We expect that as non-optical data sources are added, CSF will outperform transfer learning from ImageNet to an even greater degree. This result suggests that CSF has much to gain from adding more sensor views than we try in this work.

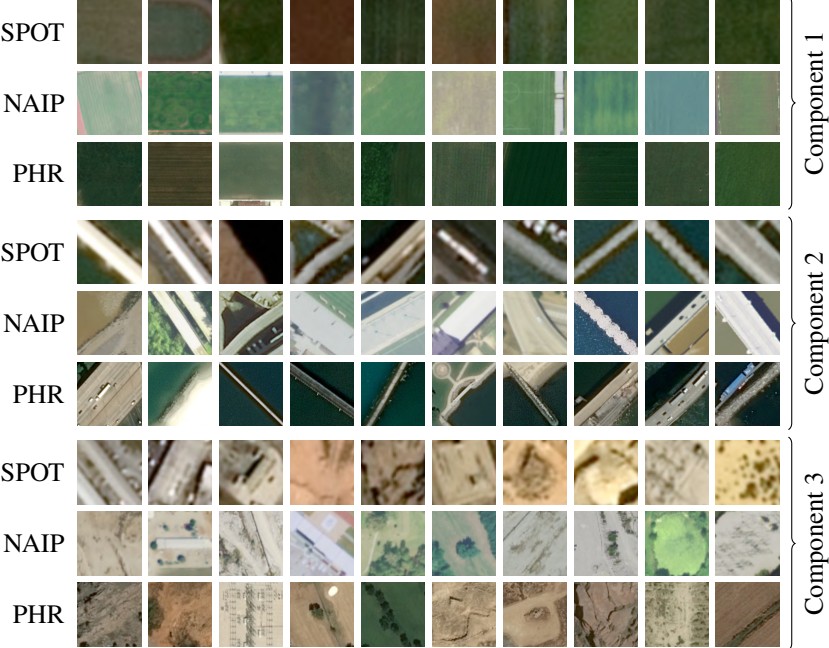

Figure 5: For each of the first three principal components of the 12-channel CSF representation space, we show 10 images from each single sensor (with inputs for the other two sensor zeroed) that maximally activate these directions. These principal components of representation space represent contain concepts (fields, bridges, and bare ground / concrete) stable across sensor combinations.

Finally, in Figure 5 we demonstrate that CSF learns high-level representations that are stable across sensors. As before, we identify the principal components of the encoder's representaitons of the 12-channel OSM dataset (which, as Szegedy et al. (2014) note, are not distinguished from individual channels). Then, we encode each sensor's views individually and visualize the images which maximally activate those linear combinations in Figure 5. We find that each direction visualized corresponds to the same high-level concept, regardless of which sensor produced it. This suggests

that CSF learns to fuse multiple sensors in early layers into disentangled and sensor-invariant features which will be easy for later layers and transfer tasks to use due to their directional consistency.

## 5 CONCLUSIONS AND FURTHER WORK

In this work we present Contrastive Sensor Fusion (CSF), a new self-supervised training objective to learn fused representations of multiple overhead image sources. CSF uses a contrastive loss to train an encoder that can produce a shared representation from any subset of available channels across multiple sensors. Using a dataset of 47 million unlabeled coterminous image triplets, we train an encoder to produce semantically meaningful representations from any possible combination of channels from the input sensors, out-performing fully-supervised ImageNet weights which required 14 million labels to train. We show through experiments that the network is successfully fusing multiple sensor information into representations that improve with additional views. We also show that, through sensor fusion, multiple low resolution sensors can outperform a single higher resolution sensor. While this work considered only optical sensors with similar resolutions, remote sensing practitioners frequently use a varity of sensors including non-optical and hyperspectral imagery with many channels. We expect multi-sensor representations of these to outperform supervised learning transferred from natural imagery to an even greater degree than demonstrated here.

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

APPENDIX

## A.1 View Creation and Loss Detail

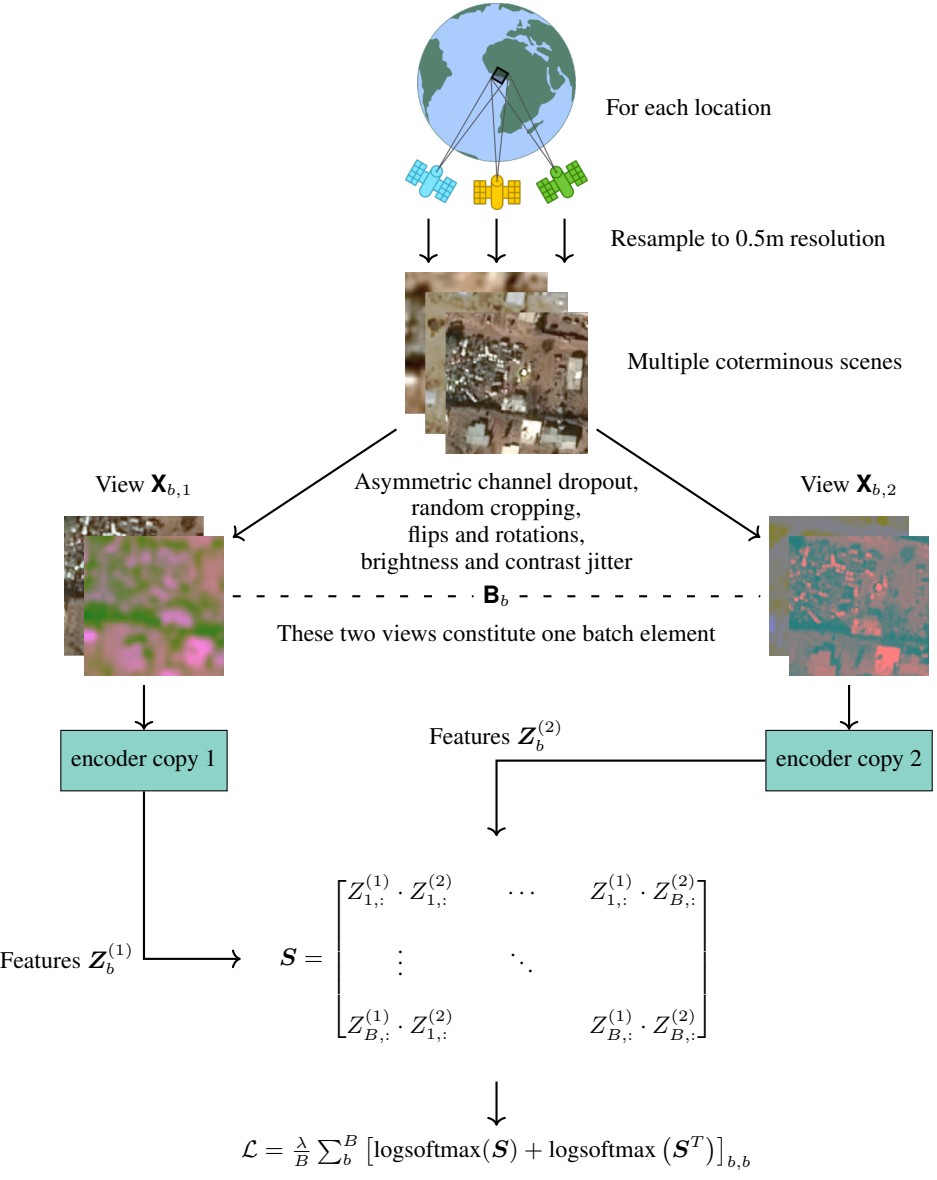

Figure A.6: This figure shows the flow of data during training, from the creation of views to the loss calculation. The loss is shown for one feature extraction block $i$. $\boldsymbol{Z}^{(1)} \times \boldsymbol{Z}^{(2)T}$ is a batchwise matrix of feature similarities; the diagonal corresponds to the similarity of representation of *matched* batch elements.

## A.2 Efficiently Computing the Contrastive Loss

The primary advantage of unsupervised training is that it grants access to much more data. To take full advantage of this, each training step must be efficient.

Using $\phi(z^{(1)}, z^{(2)}) = z^{(1)T} z^{(2)}$ and drawing negative NCE samples from other images in the batch enables computing the contrastive loss very efficiently. Given a batch of two feature maps, $\boldsymbol{Z}^{(1)}, \boldsymbol{Z}^{(2)}$

both of shape $(B, N^2)$ where $B$ is our batch size and $N^2$ is the flattened size of each spatially down-sampled feature map, we define their similarity matrix $\boldsymbol{S}$ as

$$\boldsymbol{S} = \boldsymbol{Z}^{(1)} \times \boldsymbol{Z}^{(2)^T} = \begin{bmatrix} Z^{(1)}_{1,:} \cdot Z^{(2)}_{1,:} & \cdots & Z^{(1)}_{B,:} \cdot Z^{(2)}_{1,:} \\ \vdots & \ddots & \\ Z^{(1)}_{1,:} \cdot Z^{(2)}_{B,:} & & Z^{(1)}_{B,:} \cdot Z^{(2)}_{B,:} \end{bmatrix} \tag{A.1}$$

Each element of this matrix measures, per batch, how similar (under $\phi$) elements' features generated from view 1 are to all other elements' features generated from view 2. Thus, the diagonal elements represent similarity of features from samples of the same origin but different dropout, cropping, and color and contrast jitter. We measure the neural net's ability to associate features of identical origin with one another, rather than negatives in the same batch, treating this as a classification problem. We take the log-softmax along each axis to find predicted associations from view 1 to view 2 and vice-versa.

The contrastive loss at this layer is computed as the mean of diagonal elements of the log-softmax of both the similarity matrix and its transpose

$$\mathcal{L}_{\text{contrastive}}(\boldsymbol{Z}^{(1)}, \boldsymbol{Z}^{(2)}) = \frac{1}{B} \sum_b^B \left[ \text{diag} \left( \text{log-softmax}(\boldsymbol{S}) \right) + \text{diag} \left( \text{log-softmax} \left( \boldsymbol{S}^T \right) \right) \right]_b \tag{A.2}$$

The diagonal elements of the log-softmax of the similarity matrix give the loss for predicting view 2 from view 1, and performing the same process with the similarity matrix's transpose (equivalently, the softmax along the other axis) gives the loss for predicting view 1 from view 2.

### A.3 TRAINING DATASET

Our training dataset uses imagery collected by three different sources, each with four channels (bands): red, green, blue, and near-infrared (NIR).

**SPOT** *Satellite Pour l'Observation de la Terre (Satellite for observation of Earth)*
Consists of world-wide imagery at a pansharpened resolution of 150 cm. Our imagery comes from the OneAtlas basemap published by Airbus, derived from images acquired by SPOT-6/7 satellites.

**NAIP** *National Agriculture Imagery Program*
Consists of roughly yearly aerial photography at a 100 cm resolution over each of the states in the contiguous United States. From the USDA's Farm Service Agency.

**PHR** *Pléiades High Resolution*
Data acquired from the OneAtlas basemap consisting of measurements from two satellites, Pléiades-HR 1A and Pléiades-HR 1B, at a pansharpened resolution of 50 cm and limited mostly to urban areas.

Although NAIP, for instance, is a project utilizing multiple different aircraft-based cameras, we treat them all as one data source for the purposes of this paper. We similarly treat the imagery from the SPOT and PHR satellite constellations as single data sources.

The intersection of the coverage of these satellites gives us an area of about 532,000 km$^2$, which we tile into images at a resolution of 50 cm and of size 128-by-128 pixels (64-by-64 meters). This gives 130 million possible distinct locations over the contiguous United States. We save just under 72 million of these as triplets of measurements, one image per sensor type, and randomized the collection date so that our dataset would capture persistent rather than transient features on the Earth. The upstream loss during training saturated at 47 million images, thus we stopped training and saved the model at this point.

## A.4 OPEN STREET MAP CLASSIFICATION DATASET

The OSM classification experiments use 12 labels:

1. `man_made=bridge`
2. `man_made=breakwater`
3. `building=farm`
4. `power=substation`
5. `leisure=stadium`
6. `leisure=golf_course`
7. `waterway=dam`
8. `landuse=quarry`
9. `landuse=farmland`
10. `landuse=forest`
11. `natural=water`
12. `natural=bare_rock`

All of these features are polygons. To find a location on each feature which is still representative of the label when cropped, we find the approximate pole of inaccessibility, which is the point farthest from the exterior of the shape (Agafonkin, 2016). Then, images are upsampled to a 50 cm on-the-ground resolution and further cropped to an extent of 128-by-128 pixels, for a total spatial extent of a square of side lengths of 64 m.

