# OpenReview forum: "Representation Learning for Remote Sensing: An Unsupervised Sensor Fusion Approach"
_ICLR.cc/2020/Conference — Reject_

### Official Review · AnonReviewer2 · 2019-10-16
**Official Blind Review #2**

**Rating:** 3

**Review:**

The paper proposes an unsupervised method to embed (remotely sensed) image patches such that they are clustered by high-level "image content" rather than by low-level pixel statistics. The argument is that in remote sensing one often has access only to a subset of all possible images/channels, so it would be useful to get the same embedding for each subset depicting the same scene. The basic idea is to generate, during training, two versions of the same patch by randomly dropping out channels, and to penalise the difference between the two corresponding embeddings, using the InfoNCE loss. The latter ensures that the patch remains "recognisable" among a set of random non-matching patches, so as to avoid learning a trivial embedding that is the same for any input.

The high-level goal, to train an encoder that always gives a similar encoding (and consequently similar downstream results) for a given patch, independent of the sensor channels used, is certainly appealing. It is also true that one would like to do this with unsupervised learning or transfer learning, since labelled data is scarce in remote sensing. Unfortunately, in my understanding the paper does not make much progress towards that goal.

Novelty is limited, the work is a variation of the CMC method. Yes, it splits channels randomly, not by views; and uses a Siamese architecture, not separate encoders; which it can afford to do since there are no viewpoint differences. And it computes the loss at more than one layer - which is a fairly trivial and popular thing to do. Moreover it is not well justified in this case, since all that matters in a feed-forward architecture is that the final embeddings are similar. Constraining lower layers might empirically help - in particular speed up learning - but conceptually there is no reason to reduce the lower layers' capacity to somehow turn different inputs into the same output. If they are really very different (think RADAR images) this could in principle even hurt.

Both in the discussion and in the experiments, I miss crucial and obvious baselines. First, it does in my view not make sense to use pre-trained ImageNet features as the only baseline. Yes, they are known to work surprisingly well also for remote sensing data - but, unsurprisingly, not nearly as well as pre-training on similar remote sensing images. A more natural baseline would be to pre-train on the same type of imagery, with a proxy task for which it is easy to get labels automatically  - actually the paper implicitly suggests that labels from OpenStreetMap might be a good candidate. Beyond reducing the domain gap, this also has the advantage that one is not limited to 3 channels, and that one can use an architecture suitable for the task (e.g., standard architecture with a lot of pooling are known to not perform well for semantic segmentation of remote sensing data). Labels for supervised learning might in many remote sensing tasks be scarce, but to get any old labels and pre-train a reasonable proxy task is still easy, as map data is abundant.

Perhaps even more important is another natural baseline. In spirit, the proposed method is closer to a generative embedding, since the supervision signal is not a discriminative task, but rather encourages embeddings to stay unique for each patch, and hence in principle fully decodable. Arguably, the most obvious way to achieve the same thing is a Siamese pair of auto-encoders for the two training patches, with an additional loss on the difference between the latent representations. Without that baseline it is impossible to tell whether the InfoNCE loss does a good job - in some sense it feels like making the task harder if one demands not only that a patch of water can be decoded from the embedding, but that it should be distinguishable from any other patch of water that resembles it even in raw pixel space.

The data used in the experiments is somewhat unconvincing, as it is really the opposite of diverse, multi-modal remote sensing channels. The same four R-G-B-NIR channels from sensors with not so different resolutions. The claim that the method homes in on high-level similarity and overcomes low-level radiometric differences would be much more credible if one uses short-wave infrared, thermal, perhaps even Radar images. Moreover, the gains are disappointingly small: even in the best case of 12 channels, the improvement over ImageNet pre-training is roughly from 4.5/10 correct neighbours to 5/10 correct neighbours. The even lower gains with fewer channels underline the need to have a pre-trained baseline without a massive domain gap. The classification accuracy of kNN does improve with the proposed embedding (at least for some combinations - for Pleiades RGB it is the same), but that experiment is unnatural. If one has the labels to do kNN, one might as well use them to train a proper classifier, i.e., add a couple of fully connected layers and fine-tune in a supervised fashion.

A number of technical details in the experiments remain unclear. Section 4.2 suggests the channels in Fig. 4 were added in a fixed order - why? There are many potentially viable combinations of, say, 4 channels, not just one. Furthermore, Fig. 5 suggests to me that the learned similarities are in fact not "high-level", but (up to rotation invariance) fairly low-level texture properties like "homogeneous green patch", "bright straight line on dark background", etc. I am not sure what suggests to the authors that there is much sensor invariance. Again, channels with really different wavelength would be more convincing here.

In terms of presentation, the paper repeatedly makes strong, but unsupported claims about the special properties of remote sensing images.  It makes no sense to state that standard pictures often have "one subject" - ImageNet does, but a look at standard datasets like Cityscapes or DepthInTheWild shows that it is not true in general. In the same vein, I don't see why the content changes more unpredictably at the same location across time - arguably the rate of change is much lower, because only large-scale changes impact the image, and because there aren't many dynamic occluders. And the "distributional hypothesis" certainly also holds for remote sensing images, just at different scale - otherwise people would not use all sorts of smoothness priors, super-pixel segmentations, etc. when working with them. And I do not understand why the adjacent-patch approach of CPC needs nearby patches to be similar - in my view it only requires that nearby patches are not conditionally independent, so basically any type of patterns.

A small detail: some references, while not wrong, seem rather arbitrary. E.g., (Wu 2018) is a little-known random example. Why not use earlier, more standard references - for semantic segmentation of remote sensing data with deep learning one could for example think of (Maggiori, Audebert, Marmanis, Kampffmeyer, Sherrah, ...).

Overall, while the paper brings up a valid question, it does not give a convincing answer. The justification of the method and its novelty is to some degree contrived, the dataset is ill-suited to really prove the point, and the natural baselines are missing.

**Experience Assessment:**

I have published one or two papers in this area.

**Review Assessment: Checking Correctness Of Derivations And Theory:**

I assessed the sensibility of the derivations and theory.

**Review Assessment: Checking Correctness Of Experiments:**

I assessed the sensibility of the experiments.

**Review Assessment: Thoroughness In Paper Reading:**

I read the paper at least twice and used my best judgement in assessing the paper.

---

> ### Author Response · Authors · 2019-11-15
> **Response to Reviewer #2**
>
> Anonymous reviewer #2,
>
> Thank you for writing an extensive and detailed review.
>
> > “The high-level goal, to train an encoder that always gives a similar encoding [...] independent of the sensor channels used, is certainly appealing”
>
> We are glad that you see the appeal, but disagree that we do not make progress towards the goal. Fig 5 demonstrates that directions in representation space correspond to the same concepts, regardless of input sensor. Even if “the learned similarities are not ‘high level’,” the property of sensor choice invariance is both useful and, to our knowledge, unique to our work.
>
> > “Novelty is limited, the work is a variation of the CMC method”
>
> Our main result is training a network to fuse any subset of its inputs without additional training. We do not claim dramatically improved representation quality over other contrastive unsupervised learning methods; indeed our approach is philosophically similar to CMC. Sensor choice invariance is the useful new contribution.
>
> > “it does in my view not make sense to use pre-trained ImageNet features as the only baseline. [...] A more natural baseline would be to pre-train on the same type of imagery, with a proxy task”
>
> Prior work has found contrastive multiview learning to outperform other self-supervised approaches relying on proxies (e.g. colorization).
>
> > “it feels like making the task harder if one demands not only that a patch of water can be decoded from the embedding, but that it should be distinguishable from any other patch of water that resembles it even in raw pixel space.”
>
> We’d expect that pixel-space loss would work worse than representation-space loss, as prior work (CPC) suggests. Sometimes we do penalize the model for representing patches of one class the same way, but in practice the majority of batch examples correspond to distinct classes. A main challenge of unsupervised learning is that distinct identities aren’t clearly indicated in the data.
>
> > “The data used in the experiments is [...] the opposite of diverse, multi-modal remote sensing channels.”
>
> Since our approach maximizes the mutual information across views, our initial experiments use data with high mutual information. We expect that fusing more diverse data sources will improve representation quality, since it makes the contrastive task more difficult; this is a great direction for future work.
>
> > “the gains are disappointingly small”
>
> We emphasize that our approach is trained purely unsupervised, whereas the ImageNet-trained network is trained with many labels. This experiment is a demonstration of good quality representations even when input channels are missing, not of significant improvements over supervised pre-training.
>
> > “The classification accuracy of kNN does improve with the proposed embedding [...] but that experiment is unnatural. If one has the labels to do kNN, [...] train a proper classifier”
>
> We agree that fine-tuning would be a valuable experiment, though we believe the $k$ nearest neighbors fraction does measure representation quality.
>
> > “the channels in Fig. 4 were added in a fixed order - why?”
>
> We did try other combinations of channels and found that they performed comparably. We found it easiest to visualize just a few of the possible combinations of channels. We’ll mention the other combinations we’ve tried in the paper.
>
> > “Furthermore, Fig. 5 suggests to me that the learned similarities are in fact not ‘high-level’ [...]. I am not sure what suggests to the authors that there is much sensor invariance.”
>
> Our approach is trained unsupervised, so we’d be surprised to find directions in output space corresponding to complicated, multi-part objects. Rather, we’ve learned a distributed representation which groups scenes into roughly the same categories humans do. Whether the representations are high-level is distinct from sensor-invariance. For each sensor, the same direction in output space corresponds to the same concept.
>
> > “And the ‘distributional hypothesis’ certainly also holds for remote sensing images, just at different scale”
>
> While the distributional hypothesis does hold for remote sensing images at a small scale, to learn representations of large objects using “adjacent-patch” contrastive predictive coding we’d need to use image patches of many pixels, at which point the hypothesis breaks down. We miscommunicated in saying the distributional hypothesis does not hold; we’ll fix that.
>
> > “Overall, while the paper brings up a valid question, it does not give a convincing answer.”
>
> We’d like to reiterate that our main result is learning a network that can represent arbitrary subsets of its input channels, allowing a practitioner to fuse arbitrary sensors at inference-time. We know this to be useful in our field, and our approach is a step towards learning high-quality sensor-invariant features.

---

### Official Review · AnonReviewer1 · 2019-10-24
**Official Blind Review #1**

**Rating:** 3

**Review:**

The paper proposes an approach for representation learning on remote sensing data/satellite imagery inspired by recent unsupervised contrastive multiview representation learning methods (CPC, DIM, CMC). The method relies on the InfoNCE objective to contrast two different views of the data obtained by randomly cropping image patches, color jittering, channel dropping etc. The proposed method is compared to an ImageNet pretrained network in a classification task based on OpenStreetMap (OSM) features and show to outperform the ImageNet pretrained classifier.

The paper is well-written and the method is clearly explained. To my knowledge there is little prior work on unsupervised/self-supervised learning on remote sensing/satellite data and this data suits the contrastive/multiview framework very well. In that regard I appreciate the direction explored by the paper.

Here are some questions and concerns:

- I think the paper is lacking some important details, for example how are the unsupervised and ImageNet-pretrained representations transferred? Fine-tuning, or learning a classifier on top of the frozen representation? In the case of fine-tuning, the ImageNet baseline could be extended to more channels.

- As an additional baseline, how does a network trained from scratch on the available labeled training data perform?

- The authors evaluate on a single data set, that seems to not have been used previously. To make the evaluation more solid it would be good to compare on other data sets, for example on EuroSAT [1], and with other, possibly supervised classification methods, see, e.g., [1, 2].

- Did you do ablations on augmentations used? For example, is zeroing out channels more effective than copying other channels instead?

- Both forward and backward prediction losses are used, but it seems that the loss is symmetric. Does adding the backward prediction loss really help?

Overall, I like the direction the paper is exploring, but I think it would greatly benefit from adding detail on the outlined aspects and extending the evaluation.


[1] Helber P, Bischke B, Dengel A, Borth D. Eurosat: A novel dataset and deep learning benchmark for land use and land cover classification. IEEE Journal of Selected Topics in Applied Earth Observations and Remote Sensing. 2019 Jun 14;12(7):2217-26.
[2] Kussul N, Lavreniuk M, Skakun S, Shelestov A. Deep learning classification of land cover and crop types using remote sensing data. IEEE Geoscience and Remote Sensing Letters. 2017 Mar 31;14(5):778-82.



---
Update after the rebuttal:
Thanks to the authors for their detailed response. While the authors agree that some points should be improved, no attempts were made to actually strengthen the method in a revision. In particular, I do think evaluating the proposed method on prior benchmarks that were mostly used in the context of supervised methods (such as EuroSAT) is very important. Also, I still think comparison to from scratch training is crucial. Both comparisons are common practice in prior work such as CPC, DIM, and CMC (specifically, ImageNet). I therefore do not increase my rating.

**Experience Assessment:**

I have published one or two papers in this area.

**Review Assessment: Checking Correctness Of Derivations And Theory:**

I carefully checked the derivations and theory.

**Review Assessment: Checking Correctness Of Experiments:**

I assessed the sensibility of the experiments.

**Review Assessment: Thoroughness In Paper Reading:**

I read the paper at least twice and used my best judgement in assessing the paper.

---

> ### Author Response · Authors · 2019-11-15
> **Response to reviewer #1**
>
> Anonymous reviewer #1,
>
> Thank you for your constructive review. We’re glad that you also see the need for approaches to unsupervised learning for remote sensing data, and that contrastive multiview learning is a good fit for this problem. We’d like to respond to some of your points below.
>
>  > “The paper proposes an approach for representation learning on remote sensing data/satellite imagery inspired by recent unsupervised contrastive multiview representation learning methods (CPC, DIM, CMC). The method relies on the InfoNCE objective to contrast two different views of the data obtained by randomly cropping image patches, color jittering, channel dropping etc.”
>
> We’d like to emphasize that channel dropping is not only a way to strengthen the InfoNCE loss by making the learning task more difficult, but also allows the practitioner to pick the sensors to be fused at inference-time rather than training-time, by producing a network that works with many of its inputs missing. We consider this property, which we believe is unique to our work, to be central to our results.
>
> > “I think the paper is lacking some important details, for example how are the unsupervised and ImageNet-pretrained representations transferred? Fine-tuning, or learning a classifier on top of the frozen representation? In the case of fine-tuning, the ImageNet baseline could be extended to more channels.”
>
> We think there may be some misunderstanding as to our evaluation section. In our experiments, the representations are transferred as-is, with no fine-tuning or stacking additional models. We will update our paper to clarify this point, and to reflect that experiments with fine-tuning may be valuable further work.
>
> > “As an additional baseline, how does a network trained from scratch on the available labeled training data perform?”
>
> The same network trained from scratch on the OpenStreetMap data, with access to labels, would undoubtedly produce much better representations for the task we measure (clustering points by category). However, we’re aiming to demonstrate that CSF learns representations that are useful for many tasks, especially those for which large quantities of data are not available. We’d like to point out that recent work in semi-supervised learning (e.g. [TODO: cite CPC v2 paper]) has achieved results competitive with supervised learning only by dramatically changing the network architectures between the two evaluations (in this case, from AlexNet to a very large ResNet).
>
> > “The authors evaluate on a single data set, that seems to not have been used previously. To make the evaluation more solid it would be good to compare on other data sets, for example on EuroSAT [1], and with other, possibly supervised classification methods, see, e.g., [1, 2].”
>
> Thank you for pointing this out. We agree that evaluation on more standard benchmarks could strengthen our results. However, due to the aforementioned shortage of prior work on unsupervised learning in remote sensing, we did not find a comparable unsupervised benchmark using the data sources we had available — this is part of why we have open-sourced our data. While there are numerous benchmarks for supervised classification of remote sensing data, as mentioned above we do not think that comparison to a supervised baseline would be particularly helpful.
>
>  > “Did you do ablations on augmentations used? For example, is zeroing out channels more effective than copying other channels instead?”
>
> While ablation tests on augmentation strategies would be helpful for improving the quantitative performance of our method, as mentioned above we see our main contribution as the ability to train a network to operate with missing inputs.
>
> > “Both forward and backward prediction losses are used, but it seems that the loss is symmetric. Does adding the backward prediction loss really help?”
>
> Because the two views are sampled independently, the loss is not actually symmetric. The loss is computed from a similarity matrix with row $i$ corresponding to view 1 of scene $i$ in the batch, and column $j$ corresponding to view 2 of scene $j$ in the batch — the rows and columns are not interchangeable. Using both the forward and backward loss adds supervisory signal, and it is a minor computational cost relative to that of the entire ResNet encoder.
>
> [1] Helber P, Bischke B, Dengel A, Borth D. Eurosat: A novel dataset and deep learning benchmark for land use and land cover classification. IEEE Journal of Selected Topics in Applied Earth Observations and Remote Sensing. 2019 Jun 14;12(7):2217-26.
> [2] Kussul N, Lavreniuk M, Skakun S, Shelestov A. Deep learning classification of land cover and crop types using remote sensing data. IEEE Geoscience and Remote Sensing Letters. 2017 Mar 31;14(5):778-82.

---

### Official Review · AnonReviewer4 · 2019-11-10
**Official Blind Review #4**

**Rating:** 3

**Review:**

The paper presents an approach to create unsupervised representations of remote sensing images. The essential idea is to enforce similarity between representations of multiple views obtained by subsetting channels from multiple co-terminus sensor outputs.  This is implemented by training with the InfoNCE loss on high-level features (last two layers of a ResNet 50) of two views of the same image and multiple view of other images which are passed through the same weight-shared network. The evaluation is based on a custom task of classifying OSM labels. The results show that (a) the learned representation (up to 3 chosen channels) outperform a pre-trained ImageNet fine-tuned for the classification task, (b) multiple low resolution sensors can outperform a higher resolution sensor, and (c) visualisation of images indicates that stable high-level representations across sensors.

The paper makes a worthwhile contribution in introducing contrastive methods to satellite imagery - a domain well suited for analysing how contrastive methods work and also rich in applications of sensor fusion/augmentation for remote sensing applications. The experiments indicate that this is a promising direction, and the authors have helpfully open-sourced the dataset they used. The paper, however, is short on several accounts. Primarily, the experimental evaluation is sparse in detail and rigour. In addition, the paper makes unsubstantiated claims to “argue out” certain baselines from being compared. Finally, the results indicate modest improvements on a custom task, and thus remain inconclusive.

The paper bases all experimental evaluation and conclusions thereof on the custom task of classifying 12 classes of distinctive objects on 8400 images obtained from OpenStreetMap (OSM). These include diverse categories spanning generic bodies (water, forest), specific structures (substation, bridge), similar items (farm, farmland). The paper does not discuss why this is a good task (is it challenging, is it representative of remote sensing applications, are these most frequent OSM classes). Indeed, the experimental results provide no intuition on the classification task (eg. confusion matrix is essential especially given the middling accuracy of about 0.5).

The paper makes a strong claim that the “distributional hypothesis” (of expecting that spatial or temporal similarity reflect in representation similarity) does not apply to satellite imagery because “remote sensing imagery … can change abruptly between adjacent patches”. No evidence is provided of what I believe is an unintuitive claim. Indeed, existing work [1] (as cited in the paper) uses this hypothesis to create representations for satellite images. Glaringly, having made the claim, the paper does not compare proposed approach with the representations computed in [1].  The paper also makes the claim that computing representation loss should be done with high-level features rather than individual pixels. Again, this is not substantiated as the paper does not compare the proposed methods with simple auto-encoder baselines. In the absence of these baselines, the results of this paper on a custom data-set remain inconclusive.

The evaluation leaves out several other expected experiments. A few suggested conditions for ablation tests are listed:
(a) Different augmentation across channels
(b) Different values of \lambda_L (only an extreme case of last two layers has been presented)
(c) At least one other CNN-backbone, perhaps a deeper ResNet
(d) Different orderings of introducing the channels (not only by low or high resolution ordering)
(e) Different sizes of the tiles for learning representations (curiously the paper does not mention the size of the tiles as used currently)

Finally, and more broadly, the proposed approach aims to compensate the lack of supervised labels by exploiting redundancy across multiple sensors/channels. In the chosen setup, this redundancy is very high as 4 different RGBI co-terminus sensors are chosen, thereby requiring augmenting and drop-out to simulate some variation. A more realistic or challenging setup would be required to evaluate the underlying ideas.

[1] Jean, Neal, et al. "Tile2Vec: Unsupervised representation learning for spatially distributed data." Proceedings of the AAAI Conference on Artificial Intelligence. Vol. 33. 2019.

--

Update in response to rebuttal:
The authors agreed to most of the points raised, but provided no clear suggestions in addressing them. I re-emphasise that the empirical results on a single custom dataset based on OSM is limited. The author response to this remains vague. Further none of the reasonable baselines have been compared against, a point which the authors ignored in the rebuttal. In light of this, rating remains the same.

**Experience Assessment:**

I have read many papers in this area.

**Review Assessment: Checking Correctness Of Derivations And Theory:**

I assessed the sensibility of the derivations and theory.

**Review Assessment: Checking Correctness Of Experiments:**

I carefully checked the experiments.

**Review Assessment: Thoroughness In Paper Reading:**

I read the paper thoroughly.

---

> ### Author Response · Authors · 2019-11-15
> **Response to review #4**
>
> Anonymous reviewer #4,
>
> Thanks for taking the time to write a thorough review. We’d like to respond to some parts of your review below.
>
> > “The essential idea is to enforce similarity between representations of multiple views obtained by subsetting channels from multiple co-terminus sensor outputs.  This is implemented by training with the InfoNCE loss on high-level features (last two layers of a ResNet 50) of two views of the same image and multiple view of other images which are passed through the same weight-shared network.”
>
> This is a good summary of our training procedure, but we’d like to emphasize that subsetting channels at training time serves not only to create multiple views for use with InfoNCE loss, but also trains the network to operate with channels missing at inference time.
>
> > “The results show that [ . . . ] multiple low resolution sensors can outperform a higher resolution sensor”
>
> We hadn’t emphasized this as a main result of the paper, but it is indeed interesting; we’ll update the paper to reflect this.
>
> > “The paper does not discuss why this is a good task (is it challenging, is it representative of remote sensing applications, are these most frequent OSM classes).”
>
> Thank you for pointing this out. We believe that the OSM classification task demands that a representation understand the kinds of objects and relationships useful for a broad range of remote sensing applications. We’ll add a section to the paper discussing this explicitly.
>
> > “The paper makes a strong claim that the “distributional hypothesis” (of expecting that spatial or temporal similarity reflect in representation similarity) does not apply to satellite imagery because “remote sensing imagery … can change abruptly between adjacent patches”. No evidence is provided of what I believe is an unintuitive claim. Indeed, existing work [1] (as cited in the paper) uses this hypothesis to create representations for satellite images. Glaringly, having made the claim, the paper does not compare proposed approach with the representations computed in [1].”
>
> As we mention in our response to reviewer #2, we miscommunicated by saying that the distributional hypothesis does not hold at all for this imagery. Rather, it holds at a smaller scale than is useful to learn high-level representations; nearby pixels correspond to the same object on a scale of roughly meters, which means that these correlations occupy only a few pixels for even high-resolution satellite imagery. We’ll update our paper to make this more clear.
>
> > “The paper also makes the claim that computing representation loss should be done with high-level features rather than individual pixels. Again, this is not substantiated as the paper does not compare the proposed methods with simple auto-encoder baselines.”
>
> Prior work in representation learning (such as CPC, CMC, and AMDIM) has shown that computing loss from high-level features is much more effective than using a pixel-level loss. You’re right that this section of our paper is lacking justification, though; we’ll add these references and discussion to that section.
>
> > “The evaluation leaves out several other expected experiments. A few suggested conditions for ablation tests are listed:
> (a) Different augmentation across channels
> (b) Different values of \lambda_L (only an extreme case of last two layers has been presented)
> (c) At least one other CNN-backbone, perhaps a deeper ResNet
> (d) Different orderings of introducing the channels (not only by low or high resolution ordering)
> (e) Different sizes of the tiles for learning representations (curiously the paper does not mention the size of the tiles as used currently)”
>
> We agree that these ablation tests would be helpful to understand how to get maximally discriminative features. However, as stated above, our main result is learning features which are stable across every subset of input sensors, and our experiments are targeted at demonstrating that result.
>
> > “Finally, and more broadly, the proposed approach aims to compensate the lack of supervised labels by exploiting redundancy across multiple sensors/channels. In the chosen setup, this redundancy is very high as 4 different RGBI co-terminus sensors are chosen, thereby requiring augmenting and drop-out to simulate some variation. A more realistic or challenging setup would be required to evaluate the underlying ideas.”
>
> We agree that adding more distinct sensors would be beneficial, and believe this represents an exciting direction for future research. In particular, we think that adding sensor diversity should improve representation quality. As mentioned in our response to reviewer #2, we chose data sources with high mutual information for this research because our approach relies on maximizing mutual information across views.
>
> [1] Jean, Neal, et al. "Tile2Vec: Unsupervised representation learning for spatially distributed data." Proceedings of the AAAI Conference on Artificial Intelligence. Vol. 33. 2019.

---

### Decision · Program_Chairs · 2019-12-19

**Decision:**

Reject

**Comment:**

The authors present a method for learning representations of remote sensing images from multiple views. The main ideas is to use the InfoNCE loss to learn from multiple views of the data.

The reviewers had a few concerns about this work which were not adequately addressed by the authors. I have summarised these below and would strongly recommend that the authors address these in subsequent submissions:

1) Experiments on a single dataset and a very specific task: Authors should present a more convincing argument about why the chosen dataset and task are challenging and important to demonstrate the main ideas presented in their work. Further, they should also report results on additional datasets suggested by the reviewers.
2) Comparisons with existing works: The reviewers suggested several existing works for comparison. The authors agreed that these were relevant and important but haven't done this comparison yet. Without such a comparison it is hard to evaluate the main contributions of this work.

Based on the above objections raised by the reviewers, I recommend that the paper should not be accepted.